# Suppression of Breast Cancer by Small Molecules That Block the Prolactin Receptor

**DOI:** 10.3390/cancers13112662

**Published:** 2021-05-28

**Authors:** Dana C. Borcherding, Eric R. Hugo, Sejal R. Fox, Eric M. Jacobson, Brian G. Hunt, Edward J. Merino, Nira Ben-Jonathan

**Affiliations:** 1Department of Cancer Biology, University of Cincinnati Medical Center, Cincinnati, OH 45267, USA; bdana@wustl.edu (D.C.B.); E.Hugo@medpace.com (E.R.H.); Sejal.fox@cchmc.org (S.R.F.); huntbg@mail.uc.edu (B.G.H.); 2Department of Internal Medicine, University of Cincinnati Medical Center, Cincinnati, OH 45267, USA; eric.m.jacobson.biotech@gmail.com; 3Department of Chemistry, University of Cincinnati, Cincinnati, OH 45267, USA; merinoed@ucmail.uc.edu

**Keywords:** drug discovery, small molecules, targeted therapy, prolactin receptors, breast cancer, high throughput screening

## Abstract

**Simple Summary:**

Unabated tumor growth, metastasis, and resistance to hormone therapy and/or to chemotherapy constitute serious impediments for combating breast cancer (BC). With the exception of targeted anti-HER2/neu therapy and combination therapies, there have been no radical changes in the standard of care for BC patients in the past two decades. In addition, there are only limited options for treating BC-derived brain metastases that cause high morbidity and mortality. This report describes the use of high throughput screening (HTS) for identifying novel small molecules that blocked the prolactin receptor (PRLR) and suppressed BC in a laboratory setting. These small molecules have a great potential to become effective therapeutics in patients with BC.

**Abstract:**

Prolactin (PRL) is a protein hormone which in humans is secreted by pituitary lactotrophs as well as by many normal and malignant non-pituitary sites. Many lines of evidence demonstrate that both circulating and locally produced PRL increase breast cancer (BC) growth and metastases and confer chemoresistance. Our objective was to identify and then characterize small molecules that block the tumorigenic actions of PRL in BC. We employed three cell-based assays in high throughput screening (HTS) of 51,000 small molecules and identified two small molecule inhibitors (SMIs), named SMI-1 and SMI-6. Both compounds bound to the extracellular domain (ECD) of the PRL receptor (PRLR) at 1–3 micromolar affinity and abrogated PRL-induced breast cancer cell (BCC) invasion and malignant lymphocyte proliferation. SMI-6 effectively reduced the viability of multiple BCC types, had much lower activity against various non-malignant cells, displayed high selectivity, and showed no apparent in vitro or in vivo toxicity. In athymic nude mice, SMI-6 rapidly and dramatically suppressed the growth of PRL-expressing BC xenografts. This report represents a pre-clinical phase of developing novel anti-cancer agents with the potential to become effective therapeutics in breast cancer patients.

## 1. Introduction

Approximately one in eight to ten women worldwide will be diagnosed with BC in their lifetime [1]. Persistent tumor growth, metastasis, and resistance to hormone therapy and/or to chemotherapy constitute serious impediments for combating BC. Currently, treatments of BC include surgery, radiotherapy, hormonal therapy (i.e., anti-estrogens and anti-progesterone), and chemotherapy. With the exception of targeted anti-HER2/neu therapy [2] and combination therapies, there have been no radical changes in the standard of care for BC patients in the past two decades. Moreover, there are only limited options for treating BC-derived brain metastases that cause high morbidity and mortality [3]. Identification of novel and exploitable targets for therapeutic intervention has been a long-held goal for BC research.

PRL is a multifunctional hormone secreted by the pituitary lactotrophs, where it is tonically inhibited by dopamine, and is stimulated by neuropeptides, estrogen, and some growth factors. In humans, PRL is also produced in many non-pituitary sites [4,5], where its expression is regulated by an alternative promoter [6]. Human PRL (hPRL) is a 23 kDa single chain polypeptide, comprised of 199 residues. It has three disulfide bridges which stabilize its 3-D configuration, and whose locations are conserved across species [7]. The heterogeneity of the PRL-producing cells, together with the global expression of its receptor (PRLR), define PRL as a circulating hormone and an autocrine/paracrine cytokine.

The PRLR belongs to the superfamily of non-tyrosine kinase, type I cytokine receptors that bind ligands such as growth hormone (GH), leptin, some interleukins, and erythropoietin [8]. The PRLR consists of an extracellular ligand-binding domain (ECD), a short trans-membrane domain (TM), and an intracellular domain (ICD) of variable length which is linked to signaling pathways. PRL binding to pre-dimerized receptors causes conformational changes that rapidly induce phosphorylation of the ICD and the associated Janus kinase 2 (JAK2). This is followed by phosphorylation and recruitment of signal transducers and activators of transcription (STAT) 5A and 5B to the ICD. Activated STATs homo- or hetero-dimerize, translocate to the nucleus, and bind to specific sequences on the promoters of target genes. Although the JAK2/STAT5 pathway is the principal route by which PRL activates many target genes, other signaling pathways, including MAP kinase and PI3K/Akt, are also linked to an activated PRLR [9]. In breast cancer, PRLR expression is independent of estrogen receptor (ER) expression [10].

Epidemiological studies have established the involvement of circulating PRL in breast cancer (BC). A 20-year prospective analysis of 2468 women with BC and 4021 controls found strong associations between elevated serum PRL levels and increased risk of BC, especially in women with metastases [11]. In another report, two hyperprolactinemia-inducing antipsychotics, risperidone and pimozide, prompted progression of precancerous BC lesions to cancer, while aripiprazole, which did not cause hyperprolactinemia, did not [12]. Others focused on locally produced PRL [13,14]. A study that examined several hundred breast carcinomas found high correlations between tumor PRL expression and metastases, as well as shorter patient survival [15]. Using metabolic-labeling, we reported that PRL is produced by human breast epithelium, by the surrounding adipose tissue, and by infiltrating immune cells [16]. We also found increased tumor growth and development of metastases in mice implanted with PRL-overexpressing BC xenografts [17]. In addition to stimulating BC growth, PRL increases resistance to chemotherapeutic agents such as cisplatin, doxorubicin, and taxol [18].

Suppression of pituitary PRL release by dopaminergic agonists (e.g., cabergoline) has been ineffective as a potential treatment for BC [19], likely because they do not affect the locally produced PRL. Instead, to prevent PRL actions, antagonists that interfere with PRL binding to the PRLR have been generated by mutagenesis: G129R-hPRL, Δ1-9-G129R-hPRL, and S179D-hPRL [20]. However, such antagonists, as well as antibodies against the PRLR [21,22], have several limitations. These include low therapeutic efficacy, the necessity for their delivery by injections, relatively high production costs, and an inability to cross the blood brain barrier for treating BC metastases. On the other hand, small molecules as therapeutics in BC offer advantages such as the potential for oral delivery, low immunogenicity, the prospect of penetrating the brain, ease of structural optimization by medicinal chemistry, and low production costs [23].

In this study, we employed high throughput screening (HTS) of 51,000 small molecules and identified two small molecule inhibitors (SMIs), named SMI-1 and SMI-6. Both compounds abrogated PRL-induced proliferation and invasion of breast cancer cells (BCC), as well as lymphocyte proliferation and caused no apparent in vitro and in vivo cytotoxicity. A major hindrance for studying the effect of PRL on BC xenografts in mice is that the human PRLR does not recognize mouse PRL [24]. To overcome this issue, we generated BCC clones that release PRL under the control of doxycycline, and found that SMI-6 induced rapid and dramatic suppression of PRL-induced tumor growth in athymic mice inoculated with such clones.

## 2. Results

### 2.1. Strategy for HTS

The screening strategy is summarized in Figure 1A. First, we performed in silico docking of a virtual library of small molecules to the PRLR-ECD, and 1000 compounds, predicted to interfere with PRL binding, were selected from our in-house library of 340,000 small molecules. To these, another set of 50,000 diverse compounds was added. Three PRL-dependent, cell-based assays with different sensitivities to PRL were adapted to HTS, and were sequentially used to screen the 51,000 compounds (Figure 1B). In all bioassays, cells were incubated with PRL alone, compound alone, or PRL plus compound. Compounds that suppressed cell response in the absence of PRL, presumably because of cytotoxicity, were not pursued.

### 2.2. Rationale for Using Three Sequential Bioassays

Nb2 cells are rat T lymphocytes that require PRL for survival [25]. Figure 1B shows dose-dependent stimulation of Nb2 cell proliferation in response to recombinant human PRL (rhPRL), with a lowest effective dose of ~0.05 ng/mL PRL and linearity up to 0.4 ng/mL. Such high sensitivity affords conservation of rhPRL, which is needed in large amounts for screening thousands of compounds. The very high sensitivity of Nb2 cells to PRL results from a 594 base pair deletion in the PRLR-ICD [26], making this mutant receptor unique, and not fully representative of the typical signal transduction by hPRLR.

From the initial screening of the 51,000 compounds at a fixed concentration of 10 μM, 120 candidate compounds were selected. A second bioassay using a Ba/F3 cell line, stably transfected with hPRLR, was used for dose-dependent IC50 analysis. Ba/F3 cells are mouse pro-B lymphocytes which normally depend on interleukin-3 (IL-3) for survival. When stably transfected with a hPRLR-expressing construct, hPRL can replace IL-3 to support cell growth [27]. Ba/F3 cells have similar properties as Nb2 cells, but are ~20 times less sensitive to rhPRL, with a lowest effective dose of ~1 ng/mL PRL, and linearity to 8 ng/mL (Figure 1B).

To verify that compounds identified by the hPRLR-expressing BaF3 lymphocytes are active in human BCC with endogenous PRLR, the third bioassay used T47D cells stably transfected with a luciferase reporter driven by a PRL-responsive (23X GAS element) promoter (T47D-GAS/luc). This reporter responds to ligands which activate the JAK2/STAT signaling pathway. Although T47D cells have a high density of the PRLR, they have relatively low sensitivity to rhPRL, with a lowest effective dose of ~20 ng/mL hPRL and linearity up to 125 ng/mL (Figure 1B). The low sensitivity is explained by the presence of several PRLR isoforms with shorter intracellular domains. The short PRLR isoforms have similar binding affinities to PRL as the full-length receptor, but are inhibitory to PRL signaling [28]. Consequently, relatively high concentrations of rhPRL are required to elicit a response via the long PRLR isoform in these cells.

### 2.3. Analysis of Binding Affinity of the SMIs to PRLR-ECD by Microscale Thermophoresis (MST)

Seven compounds with IC50s values in the Ba/F3 assay ranging from 0.09 to 2.07 μM (Figure 1C) were analyzed for binding to the PRLR-ECD using MST. Of the 7 compounds, SMI-1, SMI-6, and SMI-7 bound the PRLR-ECD at Kds of 1.26, 3.31, and 2.69 μM, respectively. The other four compounds did not bind the PRLR-ECD and likely act by interacting with the ICD or by inhibiting post-receptor signaling. Notably, SMI-1 was among the 1000 compounds predicted by computer modeling to interfere with PRL binding to the PRLR.

Figure 2 depicts the binding curves of PRL (Figure 2A) and SMI-1 (Figure 2B) to the PRLR-ECD, and the calculated Kds, as determined by MST. The binding curve of SMI-6 is very similar (data not shown). Receptor binding of both SMIs was also confirmed by isothermal titration calorimetry. The ~40X ratio of antagonist/PRL binding affinities (i.e., 1.26 μM vs. 29.9 nM) is well within the range of effective concentrations of small molecule inhibitors of other cytokine receptors [29].

### 2.4. Selection of Compounds for Progression

The three SMIs that bound to the PRLR-ECD were then interrogated by computer modeling for predicted solubility and for ADME (absorption, distribution, metabolism, and elimination) properties. Both SMI-1 and SMI-6 were found to have good drug-like properties and were selected for progression. A docking simulation of SMI-1 and SMI-6 to a dimerized PRLR-ECD shows potential sites of their interaction with the PRLR near a PRL binding site (Figure 2C,D). Notably, the binding site of SMI-6 is adjacent to a WSXWS motif, considered essential for activation of the PRLR-ICD by PRL [30].

### 2.5. Both SMIs Antagonize PRL-Induced Cell Invasion

Using Boyden chambers, MDA-MB-468 BCC were plated on Matrigel-coated porous membranes in inserts. The inserts were suspended over chambers with medium containing either 10% fetal bovine serum (FBS), serving as a source of chemo-attractants, SMIs, PRL, or SMIs + PRL. After 24 h, cells that invaded the underside of the membrane were counted. As shown in Figure 3A, PRL at 1 nM (23 ng/mL) was nearly as effective as FBS as an inducer of cell invasion. Either SMI at 1 μM had no attractant activity on its own, but effectively blocked the PRL-induced invasion. Cells whose PRLR was inactivated by a CRISPR/Cas 9 approach (PRLR^−/−^) responded to FBS, but not to PRL (Appendix A).

### 2.6. Both SMIs Blocked Autocrine PRL-Induced Lymphocyte Proliferation

To examine if the SMIs suppress cell proliferation that is driven by autocrine PRL, we used human Jurkat lymphocytes. These cells are derived from leukemic T-cells, and their growth is sustained by autocrine PRL [31]. Cells were pre-treated for 24 h with 1 μM of each SMI, and then with 1 nM PRL for 24 h. Proliferation was examined by the incorporation of EdU (5-ethynyl-2′-deoxyuridine) as analyzed by flow cytometry. Exogenous PRL did not increase cell proliferation, likely because of saturation of the PRLR by autocrine PRL (Figure 3B). Each SMI completely blocked EdU incorporation, presumably by antagonizing autocrine PRL. Either SMI alone caused ~15% loss of cell number.

### 2.7. The SMIs Antagonize PRL-Induced JAK2 Phosphorylation

Since JAK2 phosphorylation is the first step in the PRLR activation cascade, we used the hPRLR-transfected Ba/F3 cells to examine if the SMIs block this action of PRL. As shown in Figure 3C, and calculated by densitometry (Figure 3D), SMI-1 and SMI-6 caused 67% and 56% inhibition, respectively, of PRL-induced JAK2 phosphorylation. Full blots of phospho Jak2 and total jak2 are presented in Appendix A, respectively. 

### 2.8. Progression with SMI-6

From this point on, we decided to continue primarily with SMI-6 because: (a) unlike SMI-1, SMI-6 at all tested doses showed no in vitro toxicity (see below), (b) a large-scale synthesis and purification of SMI-6 for in vivo studies is much easier, and (c) SMI-6 was considered more amenable to potential structural modifications by medicinal chemistry.

Since hPRL and hGH have similar receptor structures and signaling pathways [32], it was critical to verify that SMI-6 does not interfere with GH actions on the GHR, its cognate receptor. Figure 4A shows that SMI-6 significantly antagonized PRL-induced STAT5 phosphorylation in MDA-MB-468 cells, but did not affect the ability of GH to activate the GHR in PRLR-deficient T47D cells (Figure 4B), which express the GHR [33].

### 2.9. SMI-6 Has High Selectivity

To examine the selectivity of SMI-6, we used the screening platform services by DiscoverX Corp (Fremont, CA). As shown in Figure 4C, when tested at multiple doses against 168 G-protein-coupled receptors (GPCRs), SMI-6 caused inhibition of only three receptors: serotonin receptor 2C, serotonin receptor 2A, and hypocretin receptor 1 at IC50 values of 3.476, 2.395, and 6.712 μM, respectively. Future studies will examine if inhibition of these receptors by SMI-6 is of any physiological consequences. We also tested SMI-6 against 468 kinases that included receptor tyrosine kinases and cytoplasmic kinases such as JAK2. SMI-6 had no activity against any of these kinases. 

### 2.10. Effects of SMI-6 on Various BCC and Normal Human Cells

The label-free, non-invasive, continuous IncuCyte imaging system was used to determine dose-dependent effects of SMI-6 on the growth and survival of six BCC with different properties: BT474 (luminal B, ER/PR/Her2+), MCF7 (luminal A, ER/PR+, Her2−), T47D (luminal A, ER/PR+, Her2−), MDA-MB-231 (Claudine low, triple negative), ZR75-1 (luminal B, ER/PR/HER2+), and MDA-MB-468 (basal, triple negative). As depicted in Figure 5A, SMI-6 at increasing doses (from 78 nM to 20 μM) inhibited the growth of all tested BCC, including to some degree, T47D cells lacking the PRLR (Figure 5B). The loss of viable MDA-MB-468 cells in response to either 1 or 10 μM SMI-6 was marginal at 48 h and was more pronounced after 96 h (Figure 5C).

We also examined if SMI-6 altered growth of non-malignant human cells. Three primary human cells: fibroblasts, keratinocytes, and mammary epithelial cells, were incubated with increasing doses of SMI-6 as above. The IC50 values for all tested malignant and non-malignant cells are shown in Figure 5D. The IC50s for the effects of SMI-6 on the BCC ranged from 0.44 to 1.68 μM, while those for normal cells were significantly higher, ranging from 4.5 to 20.4 μM. These data indicate that up to 70-fold more SMI-6 may be needed in order to suppress such cells. Since all tested cells (except for T47D PRLR^−/−^) express the PRLR, and FBS contains lactogenic hormones, the growth inhibitory activity of SMI-6 likely reflects a combined PRLR-dependent and some PRLR-independent activities.

### 2.11. Analysis of In Vitro and In Vivo Toxicity of the SMIs

The potential cytotoxicity of the SMIs was tested by the lactate dehydrogenase (LDH) cytotoxicity assay. As shown in Figure 6A, SMI-6 showed no toxicity in MDA-MB-468 cells at all tested doses, while SMI-1 had modest toxicity at 10 μM. To examine for acute in vivo toxicity, female mice were treated with single ip injections of SMI-6 at 0, 12.5, 25, or 50 mg/kg and were observed for 7 days. SMI-6 had no acute effects on body weight (Figure 6B), and did not alter food intake, water consumption, or animal behavior. 

### 2.12. Generation and Characterization of Doxycycline-Regulated hPRL-Producing MDA-MB-468 Cells

We then decided to pursue the PRLR-dependent, as well as possible PRLR-independent, actions of SMI-6. To this end, we selected MDA-MB-468 cells, which do not express autocrine PRL, and generated three clones which conditionally express hPRL under doxycycline (Doxy) control. As determined by the Nb2 bioassay (Figure 7A), cell incubation with increasing doses of Doxy caused variable PRL release, with clone 2 (D-2) releasing significantly larger amounts of PRL than clones 1 or 3; PRL was undetectable in conditioned media from the vector-transfected controls.

To verify the functionality of Doxy-regulated PRL release, vector-transfected MDA-MB-468 cells and clone 2 were incubated with Doxy (0.25 μg/mL), together with increasing doses of taxol. The dose-dependent inhibition of cell viability by taxol was partially abrogated by PRL released from clone 2 (Figure 7B). These data confirmed our previous reports that both circulating and autocrine PRL increase resistance to chemotherapeutic agents in BC [18,34]. Next, vector transfected cells and clone 2 were orthotopically implanted in athymic nude mice, and tumor growth was monitored for 50 days. Figure 7C shows that addition of Doxy to the drinking water caused a strong acceleration of tumor growth of clone 2 relative to the controls, presumably because of the actions of locally produced hPRL. Whereas control tumors reached a plateau after 4–5 weeks, those driven by autocrine hPRL continued to grow, in agreement with our previous report on the tumor-promoting activity of autocrine PRL [17].

### 2.13. SMI-6 Causes Rapid and Dramatic Suppression of Tumor Growth In Vivo

Athymic nude mice were implanted orthotopically with vector-transfected MDA-MB-468 cells or with Doxy-regulated clone 2. Addition of Doxy to the drinking water revealed that locally produced PRL generated much larger tumors than the controls (Figure 8A,B). On day 50, SMI-6 was continuously delivered to the mice at a slow rate by sc implanted Alzet pumps. SMI-6 caused a rapid and dramatic suppression of tumor growth of clone 2 (Figure 8A), whereas it had no effect on the vector-transfected controls (Figure 8B). Exposure to SMI-6 for three weeks caused no apparent changes in food and water consumption, body weight, or animal behavior, indicating lack of toxicity. Histological examination of liver, heart, lung, and kidney from the SMI-6-treated mice showed no gross morphological changes indicative of chronic toxicity.

## 3. Discussion

For the past three decades, we and others have endeavored to establish that PRL aggravates BC by accelerating tumor growth, invasion, and metastasis, as well as by increasing chemoresistance [18,35,36,37,38,39,40]. These notions, however, are not universally accepted. Some investigators have challenged the concept of the tumor-promoting actions of PRL, arguing that elevated PRL and/or high expression of the PRLR can actually serve as markers of favorable clinic-pathological parameters and better patient survival [41]. To reconcile these discrepancies, additional studies are needed so as to identify the responsiveness of specific patient subpopulations to PRL blockade. Assessments should consider interactions of PRL with other factors that affect BC such as estrogen and progesterone, the effects of PRL variants, and the relative expression of different PRLR isoforms. Notably, it has been recently reported that the intermediate isoform of the PRLR, which can hetero-dimerize with the long isoform, acts as a proto-oncogene in breast cancer [42].

The present study offers a novel approach for blocking PRLR activation in BC by means of small molecule inhibitors. Small molecule candidates are discovered using ‘libraries’ of chemicals in combination with HTS facilities. HTS affords an automated, simultaneous testing of thousands of compounds, with the ultimate goal of identifying those that are effective and specific for a given target. HTS has become the mainstay of the pharmaceutical industry, resulting in many marketed drugs. Compounds with molecular weights of 500 or below have proven especially valuable for treating many diseases, since they can be used as oral medications. Indeed, most drugs marketed today belong to this class of compounds, including an increasing number of oral medications in cancer therapeutics.

We took advantage of our expert personnel and state-of-the-art drug discovery facility to conduct large scale screening for PRLR antagonists. The sequential use of the three bioassays with complementary properties and different endpoints made the HTS both efficient and economical. This approach ensured that all potential inhibitors of the PRL signaling cascade were identified, and that ‘hits’ recognized by the more sensitive, non-human lymphocytes also blocked PRL signaling in PRLR-expressing breast cancer cells.

Our screening efforts culminated in the identification of two highly promising compounds that bound to the PRLR-ECD at high affinity. Although both SMI-1 and SMI-6 were potent inhibitors of oncogenic actions of PRL, we elected to progress with SMI-6, as it showed little to no in vitro and in vivo toxicity, and had high selectivity when tested against hundreds of receptors and kinases. Of great importance is the finding that SMI-6 blocked PRL actions but did not interfere with the ability of GH, its sister molecule, to activate the GHR. In addition, SMI-6 was effective as a suppressor of cell viability of multiple BCC with different properties, while showing much lower suppressive activity when tested with normal human cells at concentrations which were very effective with BCC. Nonetheless, future structural modifications of SMI-6 should be undertaken so as to increase its therapeutic window and minimize its potential off-target effects.

Our in vitro data (Figure 5) revealed that SMI-6 also has some PRLR-independent anti-tumorigenic properties. One potential explanation is that SMI-6 may act as an apoptotic or anti-proliferative agent. The DiscoverX screening revealed that SMI-6 did not affect any of the 468 tested kinases, which included receptor tyrosine kinases and multiple intracellular kinases. Yet, SMI-6 could act by inhibiting other cytokine family receptors, particularly those that signal via the JAK-STAT pathway and play critical roles in breast cancer progression [43]. Indeed, our docking simulations showed a strong binding of SMI-6 near a WSXWS motif which is shared with many members of the cytokine receptor superfamily.

The strikingly rapid and robust tumor collapse caused by SMI-6, raises the possibility that in addition to blocking tumor PRLR, SMI-6 is converted in vivo to a more active metabolite. Drug metabolism commonly proceeds in three phases: phase 1, where major modifications, e.g., oxidation, hydroxylation, reduction, hydrolysis, and cyclization, occur in the liver by the cytochrome P_450_ monooxygenase. Phase II primarily involves conjugations, while phase III results in additional molecular modifications. In most cases, these processes generate inactive metabolites, and/or readily excreted derivatives. In some cases, however, molecules considered as prodrugs with a lesser pharmacological activity, are converted to active drugs in vivo by enzymatic or chemical reactions [44]. Future pharmacodynamics studies could determine if SMI-6 undergoes metabolic processing that result in products having increased anti-tumorigenic activity.

This report represents an early pre-clinical phase of developing a novel anti-cancer agent that should be further optimized and improved before it can be considered as therapeutics for BC patients. Additional analyses may include: (a) a more complete characterization of the pharmaco-dynamics and metabolic stability of SMI-6, (b) a determination of its oral deliverability, (c) resolution of the exact mechanism which governs the PRLR-independent anti-tumorigenic action of SMI-6, and (d) an examination of its ability to penetrate the brain as a potential treatment for brain metastases. Even if a genuine SMI-6 does not cross the blood brain barrier, a slight structural modification (i.e., hydroxylation) may enable it to gain access to the brain. An excellent example is L-DOPA, which easily crosses the blood brain barrier in the treatment of Parkinson’s disease, while dopamine itself (which lacks the carboxyl group of L-DOPA) does not [45].

Once the additional validation and optimization steps are accomplished, we foresee that SMI-6 could provide a triple benefit to BC patients with PRL/PRLR-expressing tumors: suppression of tumor growth, inhibition of invasion/metastasis, and enhanced efficacy of standard chemotherapeutic agents. Patients could be identified as candidates for targeted PRLR therapy by conducting an immunochemical analysis of tumor biopsies for isoform-specific PRLR with presently available selective antibodies [46].

Finally, a relevant question is whether blocking the PRLR by SMI-6 would cause undesirable side effects in treated BC patients. We think not. Indeed, cabergoline, a potent suppressor of pituitary PRL release, has been chronically prescribed to thousands of patients with hyperprolactinemia with minimal ill effects [47]. On the contrary, all known adverse effects of PRL result from its overproduction, which can cause infertility in women, impotence in men, and aggravation of autoimmune diseases [4]. The only potential caveat is that women with BC who have an infant, will not produce breast milk while treated with a PRLR-blocking drug. However, breast feeding in patients with BC is clearly not a recommended practice.

## 4. Methods

### 4.1. Animals

Twelve-week-old female C57/BL6 mice (Charles River laboratories, Wilmington, MA, USA) were used for the maximum tolerated dose experiment. After acclimatization for 7–10 days, mice received a single *ip* injection (100 µL) of SMI-6 dissolved in PEG300:glycerol:water (90:8:2). Animal behavior, food/water intake, and body weight were recorded daily. Mice were euthanized on day 7 post-injection and organs and blood were collected for histological examination. Tissues were fixed in formalin and transferred to 70% ethanol. Fixed tissues were dehydrated, embedded in paraffin, sectioned at 5–6 µM, and stained with hematoxylin-eosin.

#### Mouse Xenograft Studies

Eight-week-old female athymic nu/nu mice (Charles River laboratories, Wilmington, MA, USA) were used for xenograft studies. Mice were housed four/cage in sterile cages, kept under light/dark cycles (12 h:12 h), and were acclimated for 7–10 days before the experiments.

Vector transfected (control) and doxycycline-inducible MDA-MB-468 cells (clone 2) were suspended 1:1 in PBS/Matrigel (BD biosciences, San Jose, CA, USA) and inoculated into the inguinal mammary fat pad (2 × 10^6^ cells/100 μL). After surgery, mice were provided water containing 4% sucrose with or without 1 mg/mL doxycycline-hyclate (Thermo Fisher, Pittburgh, PA, USA). Tumor dimensions were measured twice/week and tumor volume was calculated as length × width^2^ × 0.52. In half of the mice, when tumors were about 150 mm^3^ in volume (controls) or 250 mm^3^ in volume (mice inoculated with clone 2), Alzet osmotic mini-pumps (model 1004, Durect Corporation, Cupertinoo, CA, USA), were implanted sc in the dorsal neck. These pumps with a 100-μL reservoir are rated for a continuous delivery at 0.11 μL/h for 4 weeks. The pumps delivered a 37% solution of hydroxypropyl-β-cyclodextrin (HPCD) (CTD, Gainesville, FL, USA) in PBS (control), or 50 mM SMI-6-HPCD complex in PBS. After 3 weeks, animals were euthanized and the tumors were weighed.

### 4.2. Reagents

Primary antibodies were as follows: anti-PRLR, H300 (Santa Cruz, Dallas, TX, USA); anti phospho-STAT5, and anti total STAT5 (Cell Signaling, Danvers, MA, USA); anti phospho-tyrosine 4G10 (Millipore, Burlington, MA, USA). Recombinant human PRL was the generous gift from Dr. Arieh Gertler, Hebrew University, Jerusalem, Israel. SMI-1 and SMI-6 were synthesized as shown in Appendix A.

### 4.3. Cell Culture

Rat Nb2 lymphoma cells were the gift of Dr. Arthur Buckley (University of Cincinnati). Murine pro B-cell Ba/F3-PRLR cells were the gift of Dr. Arieh Gertler. Human breast cancer cell lines MDA-MB-468, MDA-MB-231, T47D, BT474, MCF-7, MDA-MB-231, ZR75-1, Jurkat human leukemia cells, and human adult mammary epithelial cells were obtained from ATCC. Primary human foreskin fibroblasts (NHDF) and adult human keratinocytes (NHEK), were obtained from PromoCell, Heidelberg, Germany. All human cell lines were authenticated per NIH guidelines by the Michigan State University RTSF Genomics Core.

### 4.4. Generation of Genetically-Modified Cell Lines

#### 4.4.1. T47D-GAS/luc Cells Have a PRL-Responsive Luciferase Reporter

The reporter was generated by the insertion of 21 GAS (gamma interferon activation site) sequence elements 5′ to the minimal CMV promoter in the pGL4.26 luciferase reporter plasmid, encoding a firefly luciferase reporter gene (Promega, Madison, WI, USA). Wildtype T47D cells were transfected and stable clones were selected for hygromycin resistance and screened for responsiveness to PRL.

#### 4.4.2. T47D PRLR^−/−^ Cells Are PRLR Null

Cells were generated by a CRISPR/Cas9-mediated gene editing using the GeneArt Nuclease Vector (Thermo Fisher, Pittburgh PA, USA) containing the guide sequence: TGTCCAGACTACATAACCGG. Transfected cells were sorted by Cas9-orange fluorescent protein expression into 96 well plates at 1 cell/well. Surviving clones were screened for ablation of the PRLR protein expression by immunoblot and functional assays.

#### 4.4.3. MDA-MB-468-CW Cells Are Doxycycline-Regulated PRL Expressing Cells

MDA-MB-468 cells were infected with a lentivirus with the tetracycline-inducible expression vector pCW57.1, a gift from David Root (Addgene plasmid # 41393, containing the full length human PRL cDNA sequence. Clones were selected for puromycin resistance and screened for minimal hPRL expression in the absence of doxycycline and maximal expression of hPRL in the presence of doxycycline. For the xenograft studies, mice were either inoculated with clone 2, or with vector transfected cells.

#### 4.4.4. Culture Conditions

Cells were maintained at 37 °C with 5% CO_2_ in RPMI1640 (T47D) or low glucose DMEM (MDA-MB-468) with 10% FBS (Invitrogen, Carlsbad, CA, USA), and 50 µg/mL Normocin (Invitrogen, Carlsbad, CA, USA). T47D cells were supplemented with 1 μM recombinant human insulin (Sigma-Aldrich), 2 mM L-glutamine, and 10 mM HEPES, (Thermo Fisher, Pittburgh PA, USA). Where indicated, cells were plated and starved for 48 h, then treated in RPMI with 2% charcoal stripped serum (CSS) and 1% ITS+ (BD Corning, Corning, NY, USA) for T47D cells or in low glucose DMEM with 2% CSS for MDA-MB-468 cells.

### 4.5. PRL Determination by Nb2 Cells

Nb2 cells were plated at 30,000 cells/well in 96 well plates and incubated for 1 day in a starvation medium containing 1% FBS, 10% gelding horse serum, and 0.1 mM β-mercaptoethanol. Cells were then incubated in a treatment medium containing 10% gelding horse serum, 0.1 mM β-ME, and increasing concentrations of rhPRL or with conditioned media from the various tested cells. After 72 h, cell viability was determined by adding resazurin (Sigma-Aldrich, St. Louis, MO, USA) at a final concentration of 50 μg/mL. After 2 h, fluorescence was determined at 530 nm excitation/590 nm emission, using a Gemini XLS microplate fluorometer (Molecular Devices, San Jose, CA, USA).

### 4.6. Small Molecule Compound Library and HTS Facilities

The University of Cincinnati Drug Discovery Center has state-of-the-art molecular modeling capabilities, a large library of small molecules, and an automated HTS system. The small molecule library consists of 340,000 compounds, and was rationally designed to include drug-like molecules with maximal structural diversity. The automated HTS is operated by PerkinElmer plate explorer and a variety of detection systems.

#### 4.6.1. In Silico Modeling

A three-dimensional representation of the UC library was screened/docked against a model of the PRLR-ECD, derived from the published X-ray crystal structure. The PRLR model was prepared by adding missing atoms and minimizing energy, first with the backbone atoms fixed, then with all atoms free, using Yasara (Yasara Biosciences, Viena, Austria) and the YAMBER force field [48]. The all-atom simulation contained explicit solvent (0.9% NaCl, pH 7.4) and was performed at STP. Side chain pKa prediction was performed using Ewald summation followed by a round of energy minimization. A virtual representation of the UC library was prepared using default settings in LigPrep version 2.5 (Schrödinger, New York, NY, USA) and the receptor site was defined to include polar interactions between the model receptor and ligand. Sitemap was used to construct the virtual receptor site, and Glide version 5.8 (Schrödinger, New York, NY, USA) was used to dock the UC library in the receptor site [49].

#### 4.6.2. Assay Adaptation for HTS

As a prerequisite for screening, the three cell-based assays were adapted for HTS. Nb2 cells were plated at 10,000 cells/well in 384 well plates and incubated with 250 pg/mL hPRL with and without the small molecule compounds. After 72 h, cell viability was determined using the CellTiter-Fluor (GF-AFC peptide) assay (Promega, Madison, WI, USA). Ba/F3-hPRLR cells were plated at 10,000 cells/well in 384-well plates and incubated with 2.5 ng/mL hPRL with or without the small molecule compounds for 72 h. Cell viability was determined with the CellTiter-Fluor fluorescent assay as above. T47D-GAS/luc cells were plated at 20,000 cells/well in 384-well plates and were treated with 90 ng/mL hPRL with and without the small molecule compounds for 24 h. Cells were then incubated with Promega Steady-Glo luciferin reagent for 10 min, and luciferase activity was determined using EnVision luminometer (PerkinElmer, Waltham, MA, USA).

### 4.7. Microscale Thermophoresis for Analysis of Receptor Binding

Interactions between the PRLR-ECD and selected small molecule inhibitors were measured using a Monolith NT.115 instrument (NanoTemper, Munich, Germany). Purified recombinant PRLR-ECD (gift of Dr. Arieh Gertler), dissolved in PBS, was fluorescently labeled with NT.647 amine-reactive dye (NanoTemper, Munich, Germany). Serial dilutions of the test ligands from 7.6 nM to 250 µM were combined with the labeled PRLR-ECD to a final protein concentration of 100 nM. Samples were loaded into glass capillaries and analysis was performed at 25 °C in 50 mM Tris (pH 7.6), 150 mM NaCl, 10 mM MgCl_2_, 0.05% Tween-20, using 95% LED power and 20–40% IR-laser power.

### 4.8. Molecular Docking Simulation

Docking analysis to model binding of PRL or the SMIs to the crystal structure of hPRLR-ECD was performed using a web-based docking simulation service. Molecular structure files were submitted to http://www.dockingserver.com/ (accessed on 1 May 2021) (Virtua Drug Ltd., Brentwood, TN, USA) for evaluation [50]. The Docking Server uses Marvin (ChemAxon, Kft, Budapest, Hungary) and MOPAC9000 (Stewart Computation Chemistry, Colorado Spring, CO, USA) for ligand set-up and AutoDock 4 (Molecular Graphics Laboratory, The Scripps Research Institute, La Jolla, CA, USA) to simulate docking of small molecule ligands to rigid protein structures.

### 4.9. Signaling Assays

STAT5 signaling was determined using the Human JAK/STAT pathway phosphorylation Array kit from RayBiotech (Peachtree Corners, GA, USA. T47D-WT or T47D PRLR^−/−^ (PRLR null) cells were plated at 5 × 10^6^ cells/150 mm plate and starved for 48 h in RPMI with 1% ITS and 4% CSS. Cells were then pre-treated for 2 h with 10 μM SMI-6 and incubated with or without 10 nM hPRL or 10 nM GH for 1 h. Cells were lysed, and 1 mg/mL of protein was used on the arrays. Differences in signal intensity was determined by densitometry, using ImageJ.

### 4.10. Cell Proliferation

Cell proliferation was measured using a label-free, non-invasive cellular confluence assay by IncuCyte Live-Cell Imaging Systems (Essen Bioscience, Ann Arbor, MI, USA). Malignant and non-malignant cells were plated overnight in a 96-well plate (3000 cells/well), and starved for 24 h in DMEM with 4% CS. After adding various concentrations of SMI-6, plates were placed in an XL-3 incubation chamber at 37 °C and were photographed using a ×10 objective. Live cell images were collected at 2-h intervals over several days. The IncuCyte analyzer provides real-time cellular confluence data based on segmentation of high-definition phase-contrast images. Cell proliferation is expressed as an increase in percentage of confluence.

### 4.11. EdU Incorporation

DNA synthesis in Jurkat cells was monitored through incorporation of EdU (5-ethynyl-2′-deoxyuridine). Briefly, log phase cells were placed in growth medium containing 5% CSS with or without PRL or SMIs, and incubated for 24 h. EdU was then added for 2 h and cells were harvested. Cells were permeabilized, and the incorporated EdU was derivatized with Alexa Fluor 488 Azide using a Click-iT EdU Flow Cytometry Assay kit (Thermo Fisher, Pittburgh, PA, USA). Cells were stained with propidium iodide, and EdU incorporation was detected and quantified by flow cytometry.

### 4.12. Immunoprecipitation and Western Blots

MDA-MB-468 cells at 80% confluence were starved for 48 h. Cells were pre-treated with 10 µM of SMI-1 or SMI-6, and then with 1 nM rhPRL for 15 min. Cells were lysed on ice for 30 min in RIPA buffer (Roche, Indianapolis, IN, USA). Protein concentrations were measured using a BCA protein assay kit (Pierce, Rockford, IL, USA). For PRLR immunoprecipitation, 500 µg of protein extract were immuneoprecipitated with 2 µg anti-PRLR antibody overnight at 4 °C. Immune complexes were captured on protein A/G agarose beads (Thermo Fisher, Pittburgh, PA, USA) and incubated for 3 h at room temperature. Washed beads were re-suspended in SDS-PAGE gel-loading buffer, boiled for 10 min, spun at 10,000 *g*, and loaded on 8% polyacrylamide gel. Resolved proteins were electrophoretically transferred to PVDF membrane. Membranes were blocked by incubation overnight at 4 °C in 5% BSA made with TBS-T (Tris buffered saline with 1% Tween 20). Blocked membranes were incubated overnight with primary antibodies in 5% BSA with TBS-T, followed by incubation with horseradish peroxidase-conjugated secondary antibodies (GE Healthcare, Chicago, IL, USA) diluted in 5% BSA/TBS-T for 1 h. Antibody products were developed using SuperSignal chemiluminescence reagents (Pierce, Rockford, IL, USA). For samples analyzed directly (non-IP) by electrophoresis, 40 μg of the protein extracts were loaded on onto an 8 or 12% SDS gel, followed by processing as above.

### 4.13. Invasion Assay

MDA-MB-468 cells were plated at 100,000 cells/well in serum-free medium (SFM) in hanging inserts with 8 μm pores membranes coated with Matrigel (Corning, NY, USA). The inserts were suspended over wells containing SFM with vehicle (DMSO) control, 1 nM PRL, and/or SMI-1 and SMI-6. Wells containing 10% FBS served as a positive control. After 24 h, Matrigel with the non-invading cells was removed, and invading cells on the underside of the membrane were stained with Hoechst fluorescent dye. The stained membranes were mounted on slides with Fluoromount-G (Southern Biotech, Birmingham, AL, USA) and cover-slipped. Photographs were taken using a Zeiss Axioplan Imaging 2 microscope at 10X magnification. The number of cells in each field was counted in a blinded fashion. Experiments included 3 inserts per treatment with 9 random fields photographed per treatment and were repeated 2–3 times.

### 4.14. Lactate Dehydrogenase (LDH) Cytotoxicity Assay

MDA-MB-468 cells were plated in growth medium at 5000 cells/well in 96-well plates. The next day, cells were incubated with 1, 5, or 10 µM of SMI-1 or SMI-6 for 24 h, and media were changed to serum-free medium. Conditioned media were collected after 2 h, and analyzed for cytotoxicity by the LDH Cytotoxicity Detection Kit (Clontech, Mountainview, CA, USA). Released LDH was detected colorimetrically via a coupled diaphorase:formazan reaction and absorbance was determined on a ThermoMax Microplate Reader (Molecular Devices, San Jose, CA, USA).

### 4.15. Statistical Analysis

Descriptive statistics and Student’s *t*-test were performed using Microsoft Excel. One-way ANOVA with Dunnett’s post-hoc analysis and nonlinear curve fitting of dose response data were performed using GraphPad Prism 5 (GraphPad, San Diego, CA, USA). Dose response was modeled using the log (inhibitor) vs. response, variable slope equation: *Y* = Bottom + (Top − Bottom)/(1 + 10^((LogIC50 − *x*) * HillSlope). Data analysis and Kd determination for Microscale Thermophoresis were done using the NanoTemper software. Cell confluent was calculated using IncuCyte 2016 software (Essen Bioscience, Ann Arbor, MI, USA).

## 5. Conclusions

This report describes the use of high throughput screening (HTS) for identifying novel small molecules that blocked the PRLR and suppressed BC in a laboratory setting. Given few additional validation studies, these small molecules have a great potential to become effective therapeutics in patients with BC. 

## Figures and Tables

**Figure 1 cancers-13-02662-f001:**
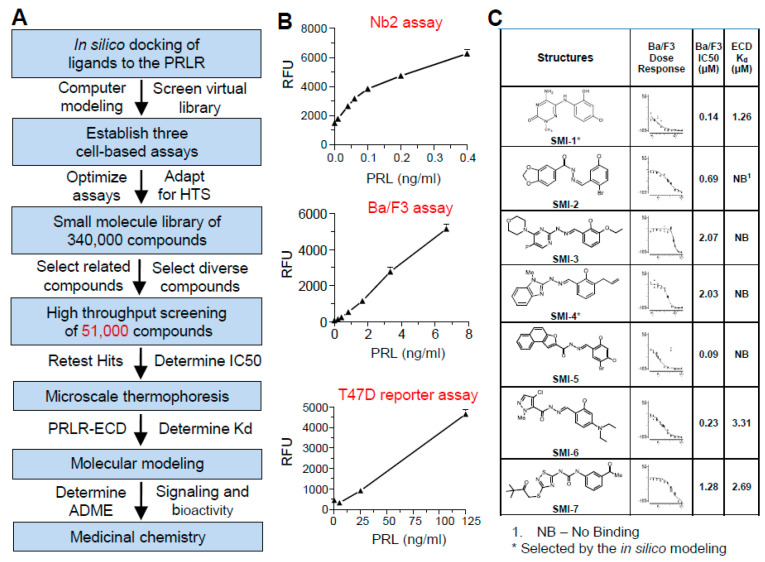
Strategy for high throughput screening (HTS) and molecular characterization of small molecule inhibitors (SMIs) of prolactin (PRL) signaling. (**A**) Steps of selecting 51,000 compounds, HTS screening, and validation. (**B**) Dose–response curves of 3 cell-based assays used sequentially. Bars are means ± SEM (*n* = 3–5). Cells in each assay were incubated with PRL alone, compound alone, or compound + PRL. RFU = relative fluorescent units. (**C**) Characteristics of the 7 SMIs with the best IC50 values in the Ba/F3 assay. Shown are the chemical structures, dose–response inhibition curves against PRL and calculated IC50s in the Ba/F3, and dissociation constants (Kd) to the extracellular domain (ECD) of the PRLR in the microscale thermophoresis (MST) assay.

**Figure 2 cancers-13-02662-f002:**
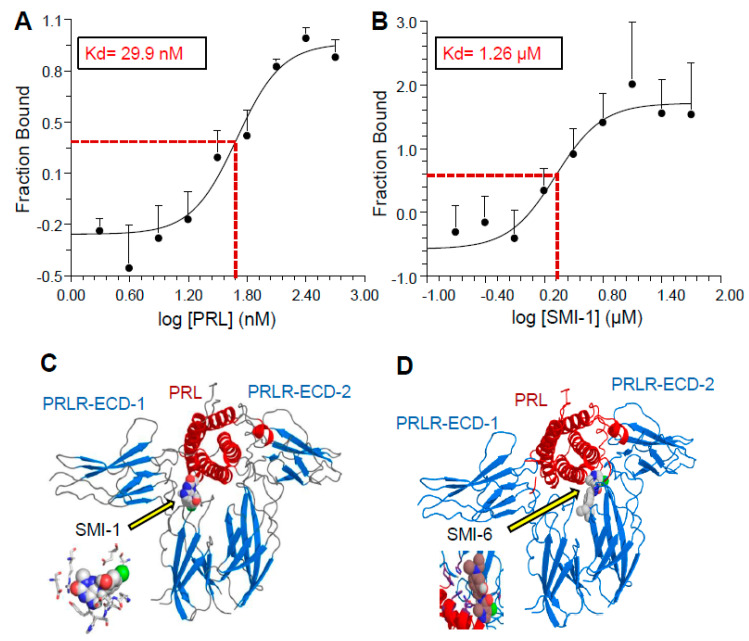
Binding curves of hPRL (**A**) and SMI-1 (**B**) to PRLR-ECD, as determined by microscale thermophoresis. The dissociation constants (Kd) were calculated using the Hill equation method. Error bars represent SEM of 3 measurements. Panels (**C**,**D**) show docking simulations of SMI-1 and SMI-6, respectively, to dimerize ECD (blue) of the PRLR bound to PRL (red). Insets show binding details of the SMIs.

**Figure 3 cancers-13-02662-f003:**
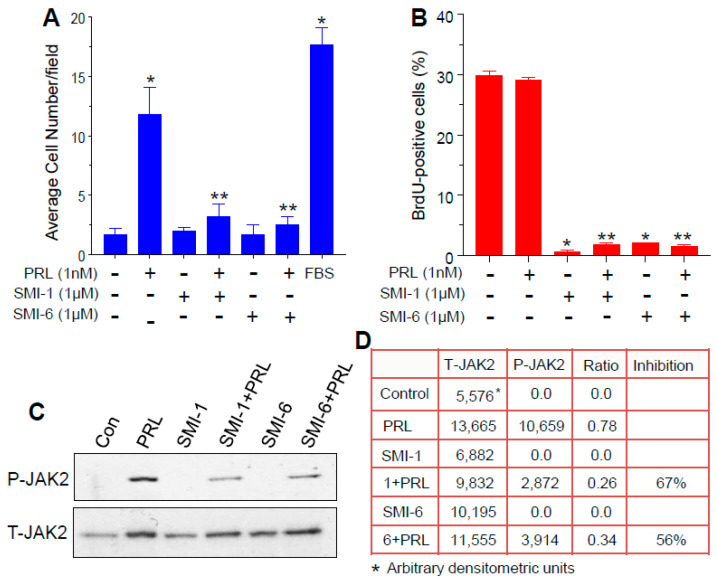
Antagonism of PRL actions by SMI-1 and SMI-6. (**A**) Inhibition of PRL-induced invasion of MDA-MB-468 cells by the SMIs, using Boyden chambers. Values are means ± SEM (*n* = 3). FBS—10% fetal bovine serum. * significant (*p* < 0.05) vs. control, ** significant (*p* < 0.05) vs. PRL. (**B**) Suppression of EdU (5-ethynyl-2′-deoxy-uridine) incorporation in Jurkat lymphocytes by the SMIs, and lack of effect of added PRL. Values are means ± SEM (*n* = 4). (**C**) Western blots showing suppression of PRL-induced JAK2 phosphorylation in hPRLR-expressing Ba/F3 by the SMIs. T-JAK2—total JAK2; P-JAK2—phosphorylated JAK2. (**D**) Densitometric quantification of P-JAK2 relative to T-JAK2 of 3 Western blot replicates.

**Figure 4 cancers-13-02662-f004:**
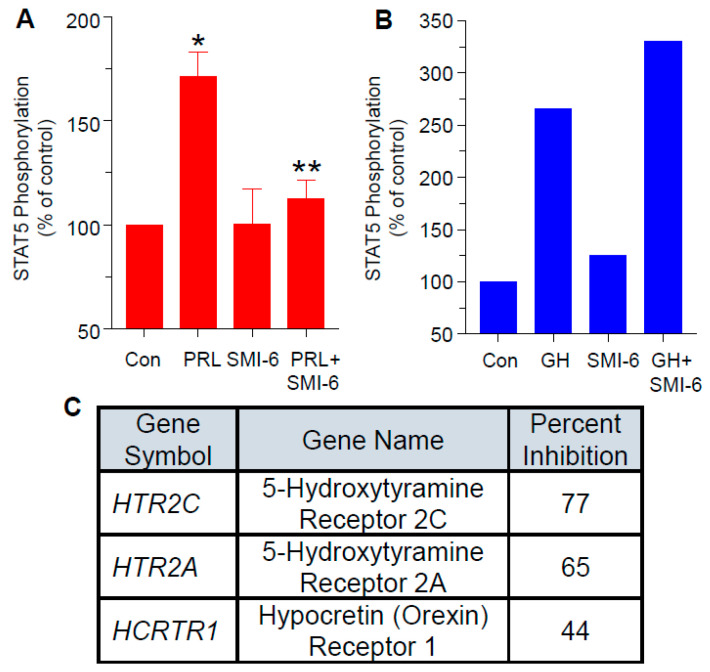
Effects of SMI-6 on induction of STAT phosphorylation by PRL or GH. MDA-MB-468 cells (**A**) or PRLR-deficient T47D cells that express the GHR (**B**) were used. Values for (**A**) are means ± SEM (*n* = 3). * significant (*p* < 0.05) vs. control, ** significant (*p* < 0.05) vs. PRL. Values for (**B**) are means (*n* = 2). PRL or GH were used at 10 nM; SMI-6 was used at 10 μM. (**C**) Results of selectivity screening of SMI-6 by DiscoverX against 168 G-protein-coupled receptors. SMI-6 caused inhibition of only 3 receptors: serotonin receptor 2C (HTR2C), serotonin receptor 2A (HTR2A), and hypocretin receptor 1 (HCRTR1). When tested against 468 kinases, SMI-6 had no activity against any kinase, including JAK2.

**Figure 5 cancers-13-02662-f005:**
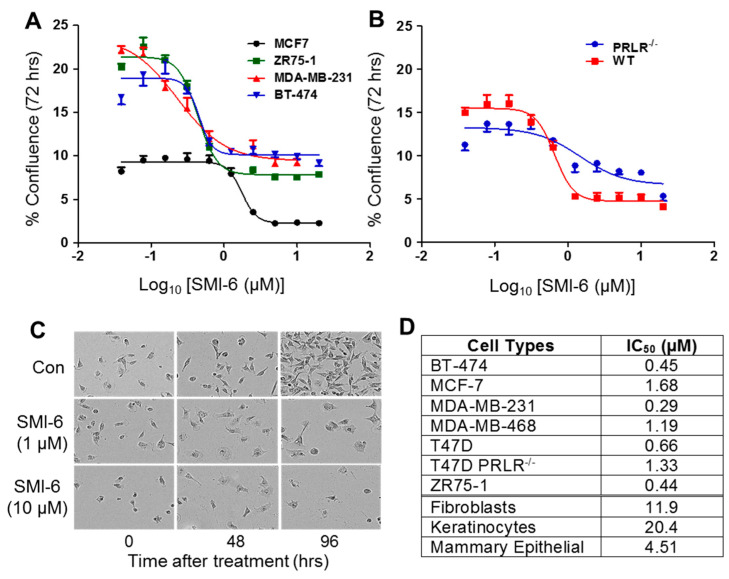
Suppression of cell proliferation by SMI-6, as determined by the IncuCyte Live-Cell Imaging Systems. (**A**) various breast cancer cells. (**B**) Wild type (WT) and PRLR-deficient (PRLR^−/−^) T47D cells. (**C**) Density of MDA-MB-468 cells treated with 0, 1, or 10 μM SMI-6 (10X). (**D**) Summary of the IC50 values of all cells analyzed for proliferation.

**Figure 6 cancers-13-02662-f006:**
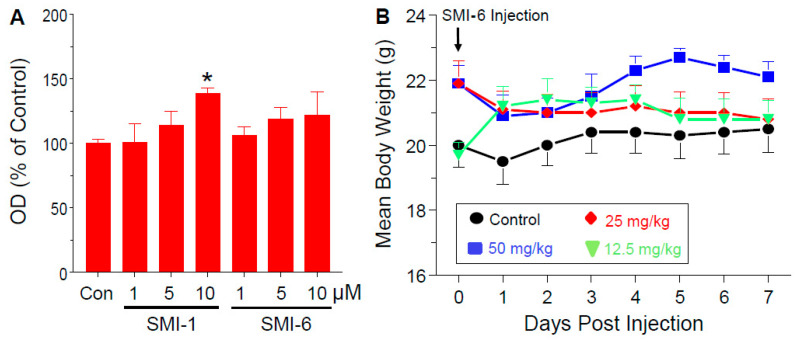
Lack of in vitro cytotoxicity or acute in vivo toxicity by SMI-6. (**A**) Lactic dehydrogenase release in response to incubating MDA-MB-468 cells with various doses of SMI-1 or SMI-6 for 24 h. Values are means ± SEM of 3 determinations. * significant (*p* < 0.05) vs. control. (**B**) SMI-6 had no effects on body weight in female mice given single ip injections of vehicle control or various doses SMI-6 and observed for 7 days. (means ± SEM; *n* = 5 mice).

**Figure 7 cancers-13-02662-f007:**
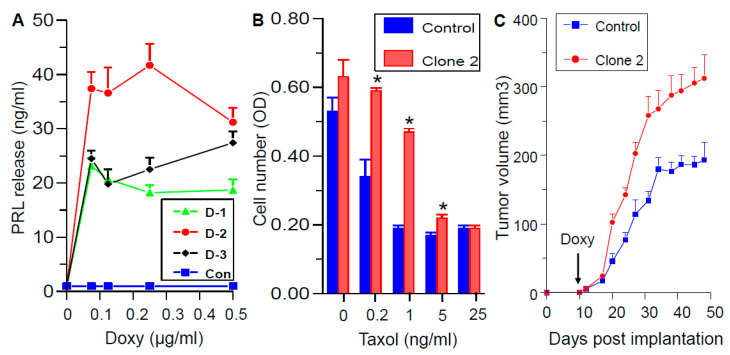
Characterization of doxycycline (Doxy)-regulated hPRL-producing MDA-MB-468 clones. (**A**) PRL release, as determined by the Nb2 bioassay, from three clones (D-1, D-2, and D-3) expressing hPRL under Doxy regulation. Cells were incubated with Doxy for 72 h. Control: vector-transfected clone. Values are means ± SEM (*n* = 3). (**B**) Autocrine PRL partially antagonizes the cytotoxic effects of Taxol. Vector-transfected clone (control) and clone 2 were incubated with 0.25 μg/mL Doxy, with and without various doses of Taxol. Cell viability was analyzed after 4 days by a Resazurin assay. Values are means ± SEM (*n* = 3). * significant (*p* < 0.05) vs. control. (**C**) Autocrine PRL accelerates tumor growth. Athymic nude mice were inoculated into the inguinal mammary fat pad with vector-transfected clone (control) or clone 2. After 10 days, doxycycline-hyclate was added to the drinking water, followed by measurements of tumor volumes for 50 days. Values are means ± SEM (*n* = 5 mice).

**Figure 8 cancers-13-02662-f008:**
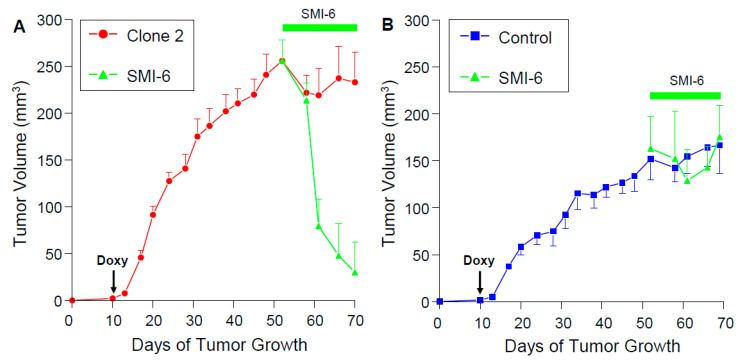
SMI-6 rapidly and dramatically suppresses growth of PRL-expressing breast cancer xenografts but does not affect control xenografts. Athymic nude mice were inoculated into the inguinal mammary fat pad with Doxy-regulated PRL-producing clone 2 (MDA-MB-468) (**A**) or with vector transfected control cells (**B**). On day 10, doxycycline-hyclate was added to the drinking water. On day 50, Alzet osmotic mini-pumps containing 50 mM SMI-6 and rated for continuous delivery at 0.11 μ/h for 4 weeks were implanted sc in the neck in half of the mice. Tumor volumes were measured throughout the experiment. Values are means ± SEM (*n* = 5–6 mice/treatment).

## Data Availability

Raw data on the selectivity screening of SMI-6 by DiscoverX Corporation can be obtained by request from the corresponding author.

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
