# Peer review of "Suppression of Breast Cancer by Small Molecules That Block the Prolactin Receptor"

_cancers, 2021, doi:10.3390/cancers13112662_

Round 1
Reviewer 1 Report
Although epidemiologic studies support a role for prolactin in the development of breast cancer, it has been more difficult to understand its role in established disease. Prolactin receptors are expressed on many clinical breast cancers, and the hormone itself is synthesized locally in breast tissue, including cancer cells. In part because of the absence of a clear role in cancer progression, potential therapeutics that target this hormone have been slow in development, and this has compounded the problem by failure to provide tools to systemically interrogate the role of prolactin, and identify markers for patients who may benefit from a targeted therapy.
In this report, the investigators used in silico screening, and three high throughput assays to screen small molecule libraries, and identify 2 lead compounds which inhibit binding of the PRL ligand to its receptor. They identified SMI-6 as a suitable lead compound, based on specificity, including failure to inhibit signaling through the closely related GH receptor, and other characteristics. This manuscript describes multiple assays to further characterize its actions, including ability to dramatically reduce growth of breast cancer cells engineered to inducibly express PRL in xenograft models. These are exciting findings, which report a novel tool with potential to enhance our understand of prolactin in this disease, and can be used for both experimental and clinical studies.
It is suggested that the manuscript could be improved by the following minor changes:
- Is PRLR expressed only on ER+ breast cancers? This was not clear—if it is also expressed on ER- cancers, this would be important.
- As noted by the authors, despite its critical role in normal mammary function, prolactin has not been easy to understand in established cancers, and accordingly, relatively little attention (and funding) have been focused on this area. In that light, a very recent paper from Clevenger’s group may also shed light on this. The lack of ability of this receptor alone or in a dimer with the full length PRLR to activate Stat5 may further explain the relative confusion in this area. https://pubmed.ncbi.nlm.nih.gov/33772010/
- Does the lack of effect of SMI-6 on tumor size of non-PRL expressing tumor cells in vivo in shown in Fig. 8B argue against an effect on immune cells or blocking angiogenesis (as suggested in the discussion, pg 11)? Or are the authors arguing that inhibition of prolactin action on the tumor cells markedly changes e.g., the cytokine environment? This requires further explanation. Note that e.g., an immune cell influx could be confirmed by immunohistochemistry.
Author Response
Thank you for your kind comments and productive suggestions.
Responses are as follows:
- With respect to PRLR and ER expressions, we added the following new sentence and a reference to this effect in the introduction (end of paragraph 3): In breast cancer, PRLR expression is independent of estrogen receptor (ER) expression [10].
- Thank you. We were not aware of the new paper the Clevenger’s group. We now added the following new sentence in the first paragraph of the discussion: Notably, it has been recently reported that the intermediate isoform of the PRLR, which can hetero-dimerize with the long isoform, acts as a proto-oncogene in breast cancer [42].
- You are absolutely correct. We must have overlooked the lack of effect of SMI-6 on non-PRL-expressing tumors upon trying to explain why SMI-6 was so active in vivo. Consequently, we have deleted the speculation on the actions of SMI-6 on either lymphocytes or on angiogenesis from the discussion, leaving only the possibility that SMI-6 may be converted in vivo to a more active metabolite.
Reviewer 2 Report
A very interesting original study identifing two potentially useful small molecule inhibitors binding prolactin receptors. The results of the paper are very useful, as these molecules could be tested in future trials as possible new drugs to treat breast cancers;
Only minor queries:
The statistical analysis section should be expanded, better explaining the various types of tests performed, and also indicating the maker and location of the statistical program used to perform the analysis.
On page 2, a short paragraph about breast cancer would be a great addition to the study; here a study you could include :DOI: 10.1007/s40264-021-01071-1
Thank You
Author Response
Thanks for your excellent comments.
Responses are as follows:
- The statistical analysis section has been greatly expanded, as you have suggested:
5.15. Statistical Analysis.
Descriptive statistics and Student’s T-test were performed using Microsoft Excel. One-way ANOVA with Dunnett’s posthoc analysis and nonlinear curve fitting of dose response data were performed using Graphpad Prism 5 (San Diego, CA). Dose response was modeled using the log (inhibitor) vs. response, variable slope equation: Y=Bottom + (Top-Bottom)/(1 + 10^((LogIC50-x)*HillSlope). Data analysis and Kd determination for Microscale Thermophoresis were done using the NanoTemper software. Cell confluent was calculated using IncuCyte 2016 software (Essenn Bioscience, Ann Arbor MI).
- Thank you. A section on breast cancer was indeed missing from the introduction. Therefore, we have transferred the first paragraph from the discussion into the introduction and it does, indeed, complement the introduction very well. We did not include, however, the reference that you have mentioned because this section appeared to be self-contained as presented.
Reviewer 3 Report
The authors have successfully demonstrated their experimental hypothesis through careful design of the experiments. Following are some comments that the authors should address.
- Add error bars for fihure-4B.
- Please add the IC50 values of SMI-6 for serotonin receptor 2C, 2A, and hypocretin receptor. This will inform the reader about the selectivity of SMI-6 to PRLR versus other receptors.
- Units for figure 5A, B should be corrected to μM from mM.
Author Response
We greatly appreciate your comments.
Responses are as follows:
- Because the effects of GH or SMI-6 on STAT5 phosphorylation in PRLR-deficient T47D cells (Figure 4B) were done in duplicate, we did not include error bar. To clarify this issue, we changed the figure legend as follows:
Figure 4. Effects of SMI-6 on induction of STAT phosphorylation by PRL or GH. MDA-MB-468 cells (A) or PRLR-deficient T47D cells that express the GHR (B) were used. Values for A are means ± SEM (n= 3). * significant (p<0.05) vs control; ** significant vs PRL. Values for B are means (N=2). PRL or GH were used at 10 nM; SMI-6 was used at 10 μM.
- Thank you. As suggested, the IC50 values for the serotonin and hypocretin receptors are now included in the Results, section 2.9: …SMI-6 caused inhibition of only three receptors: serotonin receptor 2C, serotonin receptor 2A, and hypocretin receptor 1 at IC50 values of 3.476, 2.395, and 6.712 μM, respectively.
- Thanks. The units in Figure 5A and 5B were corrected to μM.